# Rolling Bearing Fault Diagnosis via Temporal-Graph Convolutional Fusion

**DOI:** 10.3390/s25133894

**Published:** 2025-06-23

**Authors:** Fan Li, Yunfeng Li, Dongfeng Wang

**Affiliations:** 1School of Mechatronics Engineering, Henan University of Science and Technology, Luoyang 471003, China; lifan082@stu.haust.edu.cn; 2Collaborative Innovation Center of Henan Province for High-End Bearing, Luoyang 471003, China; 3Luoyang Bearing Research Institute Technology Co., Ltd., Luoyang 471033, China; 220320010052@stu.haust.edu.cn

**Keywords:** rolling bearings, variational mode decomposition, channel attention fusion layer, temporal convolutional network, graph convolutional network

## Abstract

To address the challenge of incomplete fault feature extraction in rolling bearing fault diagnosis under small-sample conditions, this paper proposes a Temporal-Graph Convolutional Fusion Network (T-GCFN). The method enhances diagnostic robustness through collaborative extraction and dynamic fusion of features from time-domain and frequency-domain branches. First, Variational Mode Decomposition (VMD) was employed to extract time-domain Intrinsic Mode Functions (IMFs). These were then input into a Temporal Convolutional Network (TCN) to capture multi-scale temporal dependencies. Simultaneously, frequency-domain features obtained via Fast Fourier Transform (FFT) were used to construct a K-Nearest Neighbors (KNN) graph, which was processed by a Graph Convolutional Network (GCN) to identify spatial correlations. Subsequently, a channel attention fusion layer was designed. This layer utilized global max pooling and average pooling to compress spatio-temporal features. A shared Multi-Layer Perceptron (MLP) then established inter-channel dependencies to generate attention weights, enhancing critical features for more complete fault information extraction. Finally, a SoftMax classifier performed end-to-end fault recognition. Experiments demonstrated that the proposed method significantly improved fault recognition accuracy under small-sample scenarios. These results validate the strong adaptability of the T-GCFN mechanism.

## 1. Introduction

Rolling bearings are ubiquitous and critical components in diverse mechanical systems. However, their operational wear and eventual failure represent significant industrial challenges, posing substantial safety hazards and incurring considerable economic losses. Notably, bearing faults are implicated in over 30% of rotating machinery failures, underscoring the urgency for effective diagnostic strategies [1,2,3]. The early detection and pre-emptive replacement of deteriorating bearings are paramount to avert catastrophic system breakdowns. Consequently, the development and implementation of robust real-time monitoring and timely maintenance protocols for rolling bearings, particularly during initial operational phases, are imperative for industrial applications.

In recent decades, vibration-based fault diagnosis for rolling bearings has advanced significantly, primarily branching into two main paradigms: signal processing techniques and artificial intelligence (AI) approaches [4]. Traditional signal processing methods typically rely on the extraction and analysis of fault-sensitive features from vibration signatures. For instance, Reference [5] introduced the Harmonic Interference Resistant Fast Kurtogram (HR-Fast-Kurtogram), enhancing the accuracy of fault feature extraction and diagnosis within short, second-scale time windows. Similarly, Reference [6] demonstrated improved diagnostic precision by integrating Variational Mode Decomposition (VMD) with wavelet thresholding for enhanced signal denoising. Despite their utility in specific contexts, these conventional methods often suffer from limitations, including a heavy dependence on domain expertise for feature engineering, susceptibility to ambient noise, and poor generalization capabilities, which restrict their applicability in complex and variable industrial environments.

Concurrently, propelled by advancements in AI and computational power, deep learning (DL) has emerged as a dominant methodology in fault diagnosis. Architectures such as Convolutional Neural Networks (CNNs), Recurrent Neural Networks (RNNs), and Graph Convolutional Networks (GCNs) have been extensively applied to bearing defect identification. Exemplifying this trend, Reference [7] presented a lightweight CNN model, achieving significant diagnostic efficiency without compromising accuracy. Reference [8] developed a parallel architecture combining CNNs and Long Short-Term Memory (LSTM) networks, validated experimentally to yield high diagnostic accuracy. Furthermore, Reference [9] proposed a sophisticated model employing a GCN-based self-attention LSTM autoencoder, demonstrating superior accuracy on benchmark bearing datasets. Nevertheless, while these DL approaches facilitate end-to-end diagnosis, their efficacy is often contingent upon large volumes of labeled training data, and they can exhibit limited robustness against noise interference and varying operational conditions, thereby constraining their practical deployment.

Addressing the prevalent challenge of inadequate fault feature extraction from limited vibration data—a common limitation of existing methods—this paper introduces a novel approach for rolling bearing fault diagnosis based on Temporal-Graph Convolutional Fusion (TGCF). Through rigorous experimental validation and comparative analysis, we demonstrate that the proposed TGCF method exhibits superior diagnostic performance and enhanced generalization capability compared to state-of-the-art techniques.

## 2. Theoretical Basis

Accurate and robust fault diagnosis of rolling bearings is paramount for maintaining industrial safety and operational efficiency. However, the inherent complexity of vibration signals, often compounded by challenges such as limited fault samples and the need for comprehensive feature extraction, poses significant hurdles for effective diagnosis. Traditional methods often struggle to capture the intricate patterns embedded within these signals. In response, advanced deep learning architectures have emerged as powerful tools due to their capability for automatic and hierarchical feature learning [10]. Specifically, Temporal Convolutional Networks excel at extracting multi-scale temporal dependencies from sequential data, while Graph Convolutional Networks are highly effective in modeling complex spatial relationships and non-Euclidean data structures. These sophisticated models, along with mechanisms for integrating diverse feature representations, form the core theoretical foundation for developing robust and intelligent fault diagnosis systems. This section provides a concise overview of these essential deep learning concepts, laying the groundwork for understanding our proposed T-GCFN.

### 2.1. Temporal Convolutional Network

The Temporal Convolutional Network (TCN) represents a specialized deep learning framework adept at processing sequential data, particularly, one-dimensional vibration signals [11]. While fundamentally derived from Convolutional Neural Networks (CNNs), TCNs incorporate crucial architectural enhancements: causal convolutions and dilated convolutions, typically implemented within residual blocks.

Causal convolutions enforce a strict temporal constraint, ensuring that the output at any given time step *t* is convolved only with inputs from time step *t* and earlier, thereby preventing any information leakage from future time steps [12]. This property is essential for modeling time-series data where future information is inherently unavailable.

Dilated convolutions are employed to exponentially increase the receptive field without a concomitant explosion in computational cost or network parameters. Unlike standard convolutions, depicted in Figure 1, dilated convolutions introduce a ‘dilation factor’, *d*, which defines a fixed spacing between the kernel taps applied to the input sequence. Figure 2 illustrates this dilation for *d* = 1; standard convolution corresponds to *d* = 0 in this specific representation, though often defined with *d* = 1 in the broader literature. Critically, *d* is typically increased exponentially with network depth (e.g., *d* = *2^l^* for layer *l*). This strategy allows the network to access a very large history (receptive field) with relatively few layers, efficiently capturing long-range temporal dependencies while mitigating the parameter burden associated with achieving similar receptive fields using standard convolutional approaches [13].

The architecture of a dilated causal convolution layer, structurally illustrated in Figure 3, synergistically combines the attributes of both dilation and causality. By incorporating a dilation factor *d* into the convolutional kernel, it expands the receptive field, enabling the capture of wider contextual dependencies without inflating the parameter count or computational overhead [14].

Given a one-dimensional input sequence $X=(x_{0},\cdots ,x_{T})\in R^{T\times d_{feat}}$, where *T* is the sequence length (time steps) and *d_feat_* represents the feature dimensionality per time step, and a filter *f* of size *n* (indexed $i\in \left\{0,\cdots ,n-1\right\}$), the dilated causal convolution output *y_t_* at time step *t* and the overall TCN receptive field *w* are formulated as follows:(1)$$F(x_{t})=\sum_{i=0}^{n-1}f(i)\cdot x_{t-d\cdot i}$$(2)$$w=(n-1)\cdot (b^{m}-1)/(b-1)+1$$
where *n* denotes the kernel size (filter length); the term t−d·i represents the index of the historical input element accessed by the *i*-th filter tap for the convolution centered at time *t*, incorporating the dilation factor *d*; *m* signifies the total number of layers in the TCN stack; and *b* is the base used for the exponential increase in the dilation factor across layers (commonly *b* =2, such that dilation *d* = *b^l^* at layer *l*).

A TCN model is typically constructed by stacking multiple residual blocks. Each residual block comprises several layers, including dilated causal convolutions and weight normalization, among other potential components. A defining characteristic is the residual connection, which adds the input to the block (*x*) to the output derived from the block’s internal transformations (*F*(*x*)). The detailed architecture of a typical residual block is illustrated in Figure 4. This residual learning framework allows the network to learn modifications (the residual *F*(*x*)) relative to the identity mapping of the block’s input. The adoption of such residual connections is crucial for effectively training deeper network architectures, alleviating potential gradient vanishing issues, and ultimately enhancing the model’s representational capacity and classification performance [15].

### 2.2. Graph Convolutional Network

Graph Convolutional Networks (GCNs) represent a class of neural networks specifically designed for processing data structured as graphs. The core principle involves generalizing the convolution operation from regular grid structures (e.g., images) to irregular graph domains. This is achieved by iteratively aggregating information from a node’s local neighborhood (including itself and its adjacent nodes) and subsequently transforming the aggregated information, thereby learning powerful node representations [16].

GCNs operate on graph data, formally defined as *G* = (*V*, *E*, *A*). Here, $V\in R^{n\times d}$ constitutes the node feature matrix, where *n* is the total number of nodes, and *d* is the dimensionality of the features associated with each node. *E* represents the set of edges connecting the nodes, and *A* is the adjacency matrix, encoding the graph’s connectivity structure.

In this study, the input graph structure is derived using the k-Nearest Neighbors (KNN) algorithm. Within this KNN graph framework, each node is connected to its *k* most similar neighbors based on their feature vectors. An illustrative example of such a KNN graph, configured with *k* = 5, is presented in Figure 5. The specific formulation for identifying the nearest neighbors for a given node *x_i_* is provided below:(3)$$\min(x_{i})=KNN(k,x_{i},\beta )$$
where *k* specifies the number of nearest neighbors to retrieve, and β denotes the sample subset (or dataset) from which neighbors are selected.

Within the constructed KNN graph, the weight *e_ij_* of the edge connecting nodes *x_i_* and *x_j_* can be computed using a Gaussian kernel, reflecting their feature similarity:(4)$$e_{ij}=\exp(-\frac{{\left\|x_{i}-x_{j}\right\|}^{2}}{2\gamma^{2}}),\;\mathrm{for}\;x_{j}\in \min(x_{i})$$
where *e_ij_* denotes the edge weight, and γ is a hyperparameter controlling the bandwidth of the Gaussian kernel.

GCN approaches are broadly categorized into spectral methods, rooted in graph signal processing, and spatial methods, which define convolutions directly on graph topology [17]. The GCN employed in this work belongs to the spectral domain category. Unlike regular grids, graph nodes exhibit variable neighborhood sizes and lack a canonical ordering, rendering standard convolution operations inapplicable. Spectral GCNs circumvent this by defining convolution in the Fourier domain associated with the graph, which is achieved using the graph Laplacian matrix [18].

For a given graph *G* = (*V*, *E*, *A*), the graph Laplacian *L* is defined as:(5)L=D−A
where *D* is the degree matrix, a diagonal matrix whose *i*-th diagonal element *D_ii_* represents the degree (number of connections) of node *i*, and *A* is the adjacency matrix, where *A_ij_* = 1 if an edge exists between node *i* and node *j*, and *A_ij_* = 0 otherwise.

The degree *D_ii_* is calculated by summing the connections for node *i* from the adjacency matrix:(6)$$D_{ii}=\sum_{j=1}^{n}A_{ij}$$

Figure 6 provides an illustrative example showcasing the degree matrix, adjacency matrix, and the resulting graph Laplacian for a simple undirected graph.

A commonly used variant is the symmetrically normalized graph Laplacian, *L_sym_*, defined as:(7)$$L_{sym}=I-D^{-1/2}AD^{-1/2}$$
where *I* is the identity matrix, *A* is the adjacency matrix, and *D* is the diagonal degree matrix.

As *L_sym_* is a real symmetric and positive semi-definite matrix, the Spectral Theorem guarantees its eigendecomposition:(8)$$L_{sym}=U\Lambda U^{T}$$
where *U* is the matrix whose columns are the orthonormal eigenvectors of *L_sym_*, and Λ is the diagonal matrix containing the corresponding real, non-negative eigenvalues ($0\leq \lambda_{i}\leq 2$).

The matrix *U* containing the eigenvectors serves as the basis for the Graph Fourier Transform (GFT). The GFT transforms a graph signal $y\in R^{n}$ (representing features or values at each node in the spatial/vertex domain) into the spectral domain $\hat{y}$ via:(9)$$\hat{y}=U^{T}y$$

Conversely, the Inverse Graph Fourier Transform (IGFT) reconstructs the spatial signal from its spectral representation:(10)$$\hat{y}=U\hat{y}$$

Conceptually, the GFT decomposes the graph signal into components associated with different graph frequencies (the eigenvalues), analogous to the classical Fourier transform. In application contexts like fault diagnosis, this transformation aims to project sensor signals onto a basis where distinct patterns (e.g., corresponding to different fault types) might be more readily separable by analyzing their spectral coefficients $\hat{y}$.

Spectral graph convolution is fundamentally defined by applying a filter $g_{\theta }$ directly to the graph eigenvalues in the spectral domain [19]. The convolution of a graph signal *x* with a spectral filter $g_{\theta }$ is determined by:(11)$$x*g_{\theta }=Ug_{\theta }(\Lambda )U^{T}x$$
where $g_{\theta }(\Lambda )$ represents the filter function $g_{\theta }$, parameterized by learnable weights θ, applied element-wise to the diagonal eigenvalue matrix Λ. However, the explicit computation of the eigendecomposition (U,Λ) required for Equation (11) is computationally prohibitive for large graphs.

To overcome this limitation, approximations based on Chebyshev polynomials were developed, which were subsequently simplified by Kipf & Welling into a highly efficient first-order approximation [20]. This approach cleverly avoids the explicit eigendecomposition. It involves introducing self-loops into the adjacency matrix and then applying symmetric normalization. This leads to the widely adopted GCN layer-wise propagation rule:(12)$$Z=\sigma (\hat{A}XW)$$
where $X\in R^{n\times d_{in}}$ is the input node feature matrix (or the output from the previous layer), with *n* nodes and *d_in_* input features per node; $W\in R^{d_{in}\times d_{out}}$ is the layer-specific trainable weight matrix, transforming features from dimension *d_in_* to *d_out_*; and $\hat{A}$$\in R^{d_{in}\times d_{out}}$ is the pre-processed adjacency matrix incorporating self-loops and symmetric normalization, effectively performing localized feature aggregation. σ(·) denotes a non-linear activation function (e.g., ReLU) applied element-wise, and $Z\in R^{d_{in}\times d_{out}}$ is the resulting output feature matrix, representing the node embeddings after the graph convolution and activation.

## 3. Proposed Temporal-Graph Convolutional Fusion Network Model

This paper introduces a novel rolling bearing fault diagnosis model based on a Temporal-Graph Convolutional Fusion Network (T-GCFN), designed to automatically extract and fuse salient fault features from rolling bearing vibration data.

The overall end-to-end workflow of the proposed T-GCFN-based fault diagnosis system, from raw signal acquisition to final fault classification, is depicted in Figure 7. Following this high-level overview, the detailed architecture of the T-GCFN model itself, illustrating its internal components and data processing within the network, is presented in Figure 8.

The process begins with the input vibration signals undergoing an overlapping sampling strategy to augment the dataset size, subsequently partitioned into training, validation, and test sets according to predefined ratios. These partitioned data segments are then pre-processed through two distinct branches: Variational Mode Decomposition (VMD) for time-domain components and the Fast Fourier Transform (FFT) for frequency-domain features.

The intrinsic mode functions (IMFs) resulting from VMD serve as the input to the TCN branch of the model. Concurrently, the frequency spectrum obtained via FFT (applied after initial cleaning; see below) is utilized to construct a k-Nearest Neighbors (KNN) graph, which forms the input for the GCN branch. These processed representations are fed simultaneously into the respective TCN and GCN modules to extract complementary temporal dynamics and graph-structural (spatial) features associated with bearing faults. The features extracted from both branches are subsequently integrated within a dedicated feature fusion module, followed by adaptive pooling (e.g., global average pooling) and a final fully connected layer to yield the diagnostic classification.

Recognizing that raw industrial vibration signals are often contaminated by significant noise and irrelevant components, which can impede effective feature extraction, we implement a preliminary denoising and normalization step prior to data partitioning and feature extraction. Specifically, the input time-domain signal segment $X=(x_{0},\cdots {,x}_{t})$ undergoes min–max normalization to produce a cleaned signal, *X_clean_*:(13)$$X_{clean}=\frac{X-X_{\min}}{X_{\max}-X_{\min}}$$
where *X_min_* and *X_max_* are the minimum and maximum values within the segment *X*, respectively, and *X_clean_* is the resulting normalized time-domain signal.

The TCN branch processes the VMD-derived components. It may initially employ a 1D convolution (with a 1 × 128 kernel), followed by a stack of TCN residual blocks. As described in the original text, four such blocks (or parallel paths) are used, potentially with exponentially increasing dilation factors (*d* = 1, 2, 4, 8) to capture multi-scale temporal dependencies, consistent with the architecture shown in Figure 4. Features extracted from these blocks are then fused (e.g., via concatenation) into a comprehensive temporal representation.

While time-domain analysis is direct, subtle fault signatures can be obscured by noise. Conversely, the frequency spectrum often exhibits more pronounced changes indicative of specific fault conditions. Therefore, we transform the cleaned time-domain signal *X_clean_* into the frequency domain using FFT:(14)$$X_{fft}=FFT(X_{clean})$$
where *X_fft_* represents the resulting frequency spectrum signal.

The GCN branch utilizes this frequency spectrum *X_fft_* as the basis for node features. A KNN graph is constructed from these spectral features (as detailed previously; cf. Figure 5 and Equation (4)), capturing relationships between signal segments based on spectral similarity. This graph is then fed into a series of GCN layers (potentially employing shared weights) for graph-based feature extraction. This process might involve graph pooling operations for dimensionality reduction, followed by standard components like Batch Normalization and ReLU activation functions.

To effectively integrate the distinct fault features extracted by the TCN and GCN branches, a dedicated feature fusion module is employed, structurally detailed in Figure 9. This module incorporates a channel attention mechanism, inspired by techniques like the Squeeze-and-Excitation block [21], designed to adaptively recalibrate channel-wise feature responses. The core idea is to explicitly model inter-channel dependencies, thereby enhancing informative features crucial for diagnosis while suppressing less relevant ones.

Specifically, the mechanism first utilizes Global Average Pooling (GAP) and Global Maximum Pooling (GMP) to independently aggregate spatial information across the input feature map $F\in R^{H\times W\times C}$ (assuming 2D features *F* resulting from the concatenation/fusion of TCN and GCN outputs; adjust dimensions if 1D). This ‘squeeze’ operation produces two distinct channel descriptor vectors: $F_{avg}\in R^{1\times 1\times C}$ and $F_{max}\in R^{1\times 1\times C}$, which encapsulate channel-wise statistics.

Subsequently, to capture complex cross-channel dependencies and generate attention weights (the ‘excitation’ step), both *F_avg_* and *F_max_* are fed into a shared Multi-Layer Perceptron (MLP). This MLP typically consists of one or more hidden layers (often incorporating ReLU activation internally), followed by an output layer. The outputs of the MLP processing of *F_avg_* and *F_max_* are then merged using element-wise summation.

The merged vector is then passed through a Sigmoid activation function (*σ*) to produce the final channel attention weights $M_{c}^{F}\in R^{1\times 1\times C}$. These weights, ranging between 0 and 1, signify the relative importance of each channel.

Finally, these attention weights $M_{c}^{F}$ are used to recalibrate the original input feature map *F* via channel-wise multiplication (broadcasting the weights along spatial dimensions), yielding the refined output features $F_{out}:F_{out}=M_{C}^{F}⊗F$. This process selectively modulates the features based on their channel-wise significance. The generation of the channel attention weights $M_{c}^{F}$ can be summarized by the following equation:(15)$$M_{C}^{F}=\sigma (MLP(GAP(F))+MLP(GMP(F)))$$
where *F* represents the input feature map to the attention module; *C* denotes the number of channels in *F*; *GAP* and *GMP* signify the Global Average Pooling and Global Maximum Pooling operations, respectively; *MLP* indicates the shared Multi-Layer Perceptron; and *σ* represents the Sigmoid activation function.

## 4. Experimental Results and Analysis

### 4.1. Experimental Dataset

The experimental validation employs the widely recognized public dataset from Case Western Reserve University (CWRU), a standard benchmark extensively utilized for research in rolling bearing fault diagnosis [22]. The dataset was acquired from an experimental test rig comprising a 1.47 kW (2 HP) induction motor, a dynamometer (serving as the load device), and the test bearings. Vibration data were captured using accelerometers mounted at various locations on the motor housing (specifically, data from the drive-end bearing sensor were used in this study, consistent with common practice).

The experiments utilized SKF-6205-2RS deep-groove ball bearings. A comprehensive dataset encompassing ten distinct bearing health states was constructed: one Normal (N) condition and nine fault conditions, comprising three fault types—Inner Race (IR), Outer Race (OR), and Ball (B)—each introduced with three different defect diameters: 0.007 inches (0.1778 mm), 0.014 inches (0.3556 mm), and 0.021 inches (0.5334 mm).

Data were collected under four distinct motor load conditions: 0 HP, 1 HP, 2 HP, and 3 HP, with vibration signals sampled at a frequency of 12 kHz. Each data sample consists of 1024 contiguous data points. To augment the dataset, samples were extracted using a sliding window approach with a 75% overlap. For each of the 10 bearing states under a specific load condition, 400 individual samples were generated. The resulting dataset was then divided using stratified random sampling into training, validation, and testing subsets with a ratio of 70%, 15%, and 15%, respectively. Details of the dataset composition are summarized in Table 1.

### 4.2. Model Training Configuration

The experiments were implemented using the Python3.10 programming language and the PyTorch2.0 deep learning framework. All model training and evaluation were conducted on a workstation equipped with an AMD Ryzen 5 4500U CPU operating at 2.38 GHz and 16 GB of RAM.

Key architectural parameters and hyperparameters for the proposed T-GCFN model are detailed in Table 2.

TCN Input: The number of intrinsic mode functions (IMFs) decomposed by VMD, serving as input to the TCN branch, was set to 4. This value was determined as optimal through preliminary analysis employing the Pearson correlation coefficient method to select the most informative modes.

GCN Input: The input features for the GCN branch were derived from the FFT spectrum of the 1024-point signal segments. Owing to the inherent conjugate symmetry of the FFT for real-valued signals, only the first half of the spectrum (representing non-redundant frequencies), corresponding to 512 data points, was utilized for constructing the graph node features. The graph structure itself was generated using the k-Nearest Neighbors (KNN) algorithm, with the number of neighbors *k* set to 10. This means each node (representing a signal segment’s spectrum) in the graph is connected to its 10 most similar neighbors based on spectral features.

Key hyperparameters influencing model performance were optimized through empirical validation. A batch size of 64 and an initial learning rate of 0.001 were selected based on these experiments. The Adam optimizer was employed for training, chosen for its efficiency and stability, attributed to its integration of momentum and adaptive learning rate capabilities, which facilitate rapid convergence. Cross-entropy loss served as the objective function, selected for its suitability in multi-class classification tasks and its capacity to provide informative gradients conducive to effective model training. The model was trained for a total of 50 epochs.

### 4.3. Experimental Results and Analysis

The diagnostic performance of the proposed T-GCFN model was evaluated primarily based on classification accuracy and loss function values. Experiments were conducted using the CWRU dataset subset (0 HP load condition) detailed in Table 1. Representative curves illustrating the training and validation loss, as well as accuracy, over the training epochs for a typical run are presented in Figure 10 and Figure 11, respectively.

As depicted in Figure 10 and Figure 11, the proposed model demonstrates remarkably stable training behavior on the CWRU dataset. Convergence is rapid, with both training and validation accuracy stabilizing at 99.91% and the corresponding loss values converging to 0.0004 within approximately the first 10 epochs. This indicates efficient learning and good generalization from the training set to the validation set.

To provide a detailed visualization of the model’s classification performance on unseen data, the confusion matrix for the test set predictions is presented in Figure 12. In this matrix, each row corresponds to the predicted class labels, while each column represents the true class labels. The diagonal elements quantify the number of correctly classified instances for each bearing state. The confusion matrix clearly indicates that the proposed model achieves exceptionally high accuracy across all 10 rolling bearing health states in the test set, with nearly all instances correctly assigned to their respective classes, demonstrating robust diagnostic capability.

### 4.4. Experimental Results and Analysis in Small-Sample Scenarios

In practical industrial settings, acquiring sufficient fault data is often challenging, as machinery typically operates under normal conditions for extended periods. This data scarcity necessitates models that perform robustly, even when trained on limited samples.

To rigorously evaluate the proposed T-GCFN model’s performance under such constraints, we simulated small-sample scenarios by creating subsets of the original training data containing 10%, 30%, 50%, 70%, 90%, and 100% of the full training samples (while keeping the validation and test sets complete). In this study, ‘small-sample conditions’ specifically refer to scenarios where the amount of available training data per fault type is significantly limited. For the CWRU dataset used in our experiments, 10% of the original training samples translates to 28 samples per fault class. This quantity is considered a challenging small sample size in rolling bearing fault diagnosis, as it severely restricts the model’s ability to learn robust and generalizable features from abundant data, thereby increasing the risk of overfitting and poor generalization in real-world applications.

Regarding the selection of this specific minimum percentage, while even smaller proportions, such as 5% (corresponding to 14 samples per fault type), could theoretically represent more extreme data scarcity: our preliminary investigations indicated that 10% already presents substantial challenges for deep learning models to achieve reliable convergence and meaningful diagnostic performance. This choice allows for a rigorous assessment of our T-GCFN’s ability to extract discriminative features from limited data, which is a primary focus of this paper, while ensuring that the dataset remains sufficiently representative for model training and comparative analysis.

#### 4.4.1. Ablation Experiment

An ablation study was conducted to validate the contribution and necessity of the key components within the proposed T-GCFN architecture. The following six model variants were compared:

Model 1: The proposed T-GCFN model (full architecture).

Model 2: T-GCFN variant using simple feature concatenation instead of the channel attention fusion module.

Model 3: Baseline TCN model using only the raw 1D vibration signal as input.

Model 4: Multi-scale TCN model processing VMD components (akin to the TCN branch of T-GCFN).

Model 5: Baseline GCN model using a KNN graph constructed from time-domain features.

Model 6: GCN model using a KNN graph constructed from FFT frequency-domain features (akin to the GCN branch of T-GCFN).

To ensure stability and mitigate random variations, each experiment was repeated five times, and the average accuracy across these runs is reported as the final result. The average test accuracies of these models on the CWRU 0 HP dataset, under varying training data proportions, are presented in Figure 13. The x-axis represents the percentage of training data used, and the y-axis represents the average test accuracy.

As illustrated in Figure 13, the proposed T-GCFN (Model 1) consistently outperforms all other ablation variants across all tested data proportions, underscoring the effectiveness and synergistic contribution of its integrated components.

Analysis of Pre-processing and Branch Architectures (Model 3 vs. 4; Model 5 vs. 6): Comparing Model 4 (VMD + Multi-scale TCN) with Model 3 (basic TCN), a significant accuracy improvement (e.g., 5.61% at the 10% data level) is observed. This highlights the benefit of VMD pre-processing and the multi-scale TCN architecture for extracting informative temporal features, especially under data scarcity. Similarly, comparing Model 6 (GCN on FFT features) with Model 5 (GCN on time-domain features) reveals a substantial accuracy gain (approx. 10% at the 10% data level) when using frequency-domain inputs. This confirms the suitability of spectral features for graph-based fault representation via GCN. Overall, Models 4 and 6 consistently outperform their respective baselines (Models 3 and 5), further validating the effectiveness of the chosen VMD, FFT, and multi-scale TCN strategies.

Analysis of Dual-Branch Structure (Model 2 vs. 4 and 6): Comparing the dual-branch model with concatenation (Model 2) against the single-branch models (Model 4 and Model 6), Model 2 generally achieves higher accuracy across different data levels. Notably, as the training data size decreases, the performance degradation of Model 2 is less pronounced compared to Models 4 and 6, maintaining accuracy above 80% even at the 10% level. This suggests that the parallel feature extraction from both temporal (VMD + TCN) and spectral (FFT + GCN) domains enhances robustness in low-data regimes.

Analysis of Fusion Mechanism (Model 1 vs. 2): This comparative analysis highlights the key technical innovation of the T-GCFN model: its adaptive channel attention fusion mechanism. Comparing the full T-GCFN model with channel attention (Model 1) against the variant using simple concatenation (Model 2), Model 1 consistently achieves superior accuracy, particularly evident with the full dataset (99.91%). Crucially, under severe data scarcity (e.g., 10% data), Model 1 maintains accuracy above 90%, whereas Model 2 experiences a more significant drop. This notable performance gap suggests that simple concatenation might lead to information loss or suboptimal feature integration. In contrast, the proposed channel attention mechanism effectively and intelligently fuses the complementary features from the two branches, dynamically weighting their contributions to enhance salient diagnostic information. This adaptive integration capability is fundamental to significantly boosting diagnostic performance, especially in challenging small-sample scenarios.

To further investigate the complexity and computational cost of the models, Table 3 summarizes the average accuracy, parameter quantity, FLOPs, and training time for all models at the 10% dataset level (corresponding to 400 samples).

As shown in Table 3, Model 1 (the proposed T-GCFN) achieves the highest average accuracy of 93.21%. Compared to Model 2 (T-GCFN with simple concatenation), Model 1 improves accuracy by 9.03% (93.21% vs. 84.18%), with a modest increase in parameters (0.99 × 10^5^), FLOPs (0.02 G) and training time (18 s). This demonstrates the effectiveness of the channel attention fusion mechanism in significantly boosting performance despite a slight increase in computational cost. Furthermore, Model 1 significantly outperforms single-branch models (Model 3–6) in accuracy, highlighting the necessity of the dual-branch architecture for robust fault diagnosis.

Despite its enhanced performance, Model 1 remains relatively lightweight with 4.11 × 10^5^ parameters and 0.12 GFLOPs. While its training time (120 s) is longer than simpler models, the substantial gain in diagnostic accuracy, especially under small-sample conditions, justifies this. This balance makes T-GCFN a practical solution for online monitoring, where precision is paramount.

#### 4.4.2. Cross-Load Condition Experiment

To further assess the model’s generalization capability and robustness under varying operating conditions (cross-load)—a critical requirement for real-world deployment—we conducted cross-load diagnostic experiments in the small-sample setting. Specifically, models were trained using data subsets (10% to 100%, as described above) from the 1 HP load condition and subsequently tested on the full dataset from the 2 HP load condition. The performance of the proposed T-GCFN (Model 1) was compared against five other representative deep learning models commonly used in fault diagnosis:

Model 1: The proposed T-GCFN model.

Model 2: Lightweight CNN [7].

Model 3: Transformer [23].

Model 4: CNN-GRU [24].

Model 5: CNN-LSTM [8].

Model 6: GCN-LSTM [9].

Again, average results over five independent runs are reported. The average test accuracies under these cross-load, small-sample conditions are presented in Figure 14, where the x-axis denotes the model type, and the y-axis represents the average accuracy achieved with varying proportions of 1 HP training data.

As shown in Figure 14, while the diagnostic accuracy of all models generally improves as the proportion of training data increases, the proposed T-GCFN model (Model 1) consistently achieves the highest accuracy across all training data proportions compared to the other five baseline models under this cross-load scenario. At the most challenging 10% data level, the Lightweight CNN (Model 2) and Transformer (Model 3) exhibit the lowest accuracies (85.24% and 86.31%, respectively), suggesting difficulty in extracting generalizable features under combined data scarcity and domain shift (load variation). Models 4, 5, and 6 (CNN-GRU, CNN-LSTM, GCN-LSTM) maintain respectable accuracies around 90% at the 10% level, but their performance varies more across different data proportions. In contrast, the proposed T-GCFN model demonstrates superior robustness; its accuracy degradation from 100% data to 10% data is minimal (only a 6.79% drop), consistently staying well above 90%. This highlights its strong generalization capability and resilience to variations in operating conditions, even when trained on limited data. In summary, these results demonstrate that the proposed T-GCFN model exhibits significantly better diagnostic performance and robustness compared to existing methods under challenging cross-load, small-sample scenarios.

#### 4.4.3. High-Noise Condition Experiments

To rigorously evaluate the robustness and superiority of the proposed T-GCFN (Model 1) under noisy conditions, we conducted extensive comparative experiments against the comparative models introduced in Section 4.4.2. These experiments were performed across various Signal-to-Noise Ratio (SNR) levels, mimicking realistic industrial environments with varying noise interference. The diagnostic accuracies of these models across varying SNR levels for the 10% sample size, representing a challenging small-sample scenario, are presented in Figure 15.

As evident from Figure 15, the proposed T-GCFN (Model 1) consistently demonstrates superior diagnostic performance compared to all comparative models across the tested noise levels. For instance, even under the most challenging condition of SNR of 2 dB, T-GCFN (Model 1) achieves an accuracy of 86.48%, significantly outperforming the best comparative model at this SNR, GCN-LSTM (Model 6), which only reached 80.32%. This represents an improvement of 6.16 percentage points in accuracy. As the SNR increases, all models show an improvement in accuracy, but the performance gap between T-GCFN and other models remains substantial, particularly at lower SNR values.

This remarkable robustness of T-GCFN can be attributed to its dual-branch architecture and the adaptive channel attention fusion mechanism, which enable the model to extract more discriminative and noise-resilient features from both temporal and graph-structural domains. These comprehensive comparative results unequivocally prove the superior robustness and generalization capability of the T-GCFN model for intelligent fault diagnosis in challenging noisy and data-scarce industrial environments.

### 4.5. Generalization Experiment on the Jiangnan University Bearing Dataset

To further evaluate the generalization capability of the proposed T-GCFN model across different types of rolling bearings and operating conditions, experiments were conducted using a publicly available dataset from Jiangnan University (JNU) [25]. This dataset features two distinct bearing types, N205 (cylindrical roller bearing) and NU205 (cylindrical roller bearing), under four health states: Normal (N), Inner Race fault (IR), Rolling Element fault (RE), and Outer Race fault (OR). Faults were artificially introduced using wire cutting. Vibration signals were acquired at a sampling frequency of 50 kHz under three different operating speeds: 600 r/min, 800 r/min, and 1000 r/min.

For each health state under each operating condition, 800 samples were generated, resulting in a total of 3200 samples per condition. The data segment length, pre-processing methods (VMD, FFT, normalization), and T-GCFN model parameters were kept identical to those used for the CWRU dataset experiments. This procedure resulted in three distinct datasets, denoted Dataset A (600 r/min), Dataset B (800 r/min), and Dataset C (1000 r/min), corresponding to the three operating speeds. The composition of these datasets is detailed in Table 4.

Experiments were performed independently for each of the three operating conditions (Datasets A, B, and C). The diagnostic accuracies achieved by the T-GCFN model are presented in Table 5.

As shown in Table 5, the T-GCFN model exhibits a slight decrease in accuracy on the JNU dataset compared to the CWRU dataset. This minor reduction might potentially be attributed to factors such as signal distortion caused by interference during data acquisition or transmission specific to the JNU setup. Nevertheless, the model achieves a high overall average accuracy of 99.59% across the three different operating conditions. This consistently strong performance further substantiates the excellent generalization ability and robustness of the proposed T-GCFN model for rolling bearing fault diagnosis across varying bearing types and operational speeds.

## 5. Conclusions

Addressing the challenges of incomplete feature information, sensitivity to operating conditions, and noise interference inherent in rolling bearing fault diagnosis, particularly under small-sample scenarios, this paper proposed a novel diagnostic method based on a T-GCFN. Comprehensive experimental validation confirmed the superiority of the proposed approach, leading to the following key conclusions:(1)The T-GCFN model introduces a novel dual-branch architecture that synergistically extracts complementary temporal and graph-structural fault features from raw signals. Its core innovation lies in the adaptive fusion of multi-domain features via a channel attention mechanism, which dynamically weights and integrates information from both TCN and GCN branches. This unique design effectively addresses the challenge of incomplete feature extraction by enhancing salient diagnostic information.(2)Extensive experiments were conducted on the CWRU benchmark dataset under simulated small-sample conditions, encompassing constant load, cross-load variations, and high-noise environments. Ablation studies systematically validated the rational design and contribution of each component within the T-GCFN model. Under challenging cross-load conditions, the proposed T-GCFN consistently outperformed several state-of-the-art deep learning models, including Lightweight CNN, Transformer, CNN-GRU, CNN-LSTM, and GCN-LSTM, demonstrating superior generalization. Furthermore, the model exhibited strong robustness against high levels of noise. Its generalization capability was further substantiated through successful application to the distinct Jiangnan University (JNU) bearing dataset.(3)Furthermore, the T-GCFN model exhibits promising potential for real-time system monitoring. Its efficient inference capabilities, coupled with its end-to-end learning paradigm, make it suitable for rapid and automated fault detection in dynamic industrial settings. The computational efficiency of its core components (TCN, GCN) and pre-processing steps (VMD, FFT) further supports its applicability for online diagnostic tasks.

In summary, the proposed T-GCFN fault diagnosis method demonstrates enhanced and comprehensive feature extraction capabilities, achieves higher diagnostic accuracy, and shows strong potential for real-time deployment compared to existing approaches, especially under challenging conditions like data scarcity, varying loads, and noise interference. However, the current study primarily focused on single fault types. The applicability and performance of the T-GCFN model for diagnosing compound faults (e.g., simultaneous inner race cracking and roller spalling) warrant further investigation in future work.

## Figures and Tables

**Figure 1 sensors-25-03894-f001:**
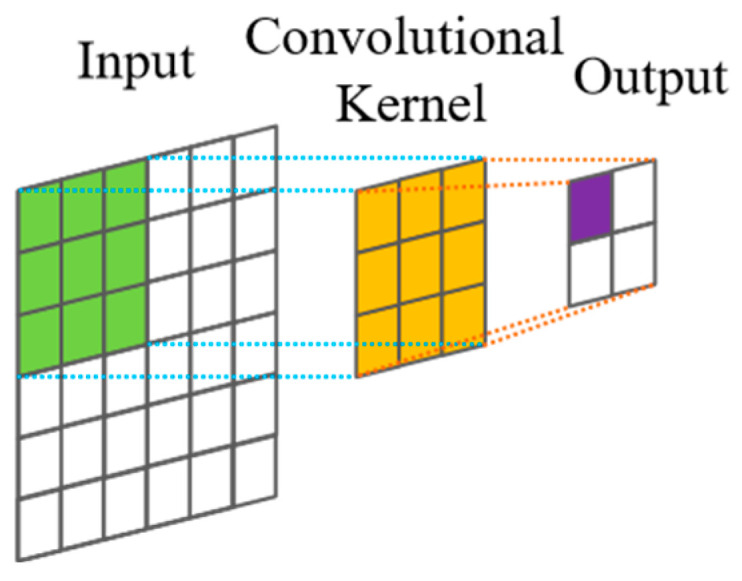
Standard convolution.

**Figure 2 sensors-25-03894-f002:**
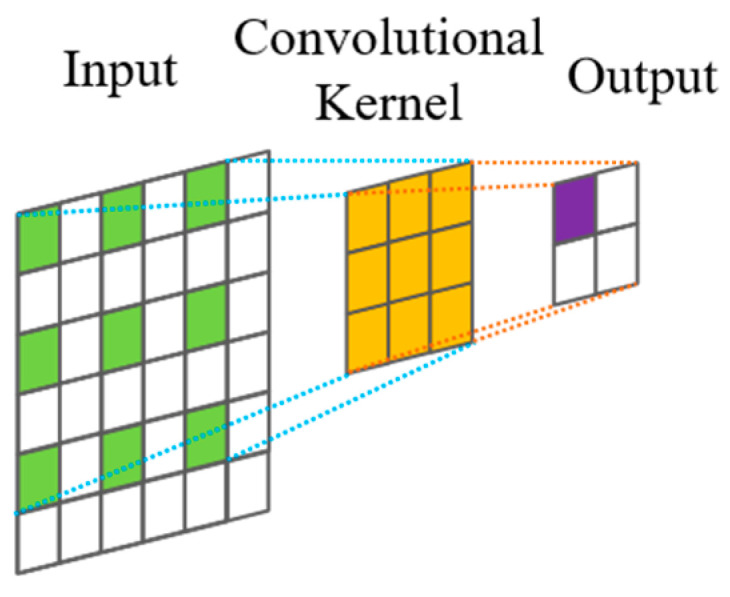
Dilated convolution.

**Figure 3 sensors-25-03894-f003:**
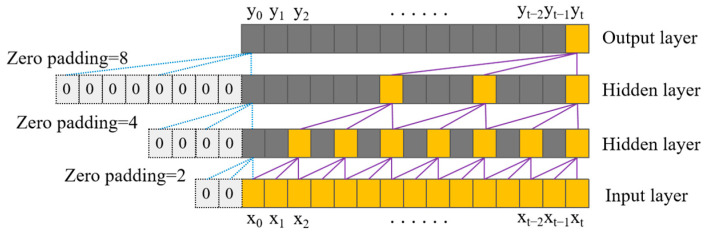
Structure of dilated causal convolution.

**Figure 4 sensors-25-03894-f004:**
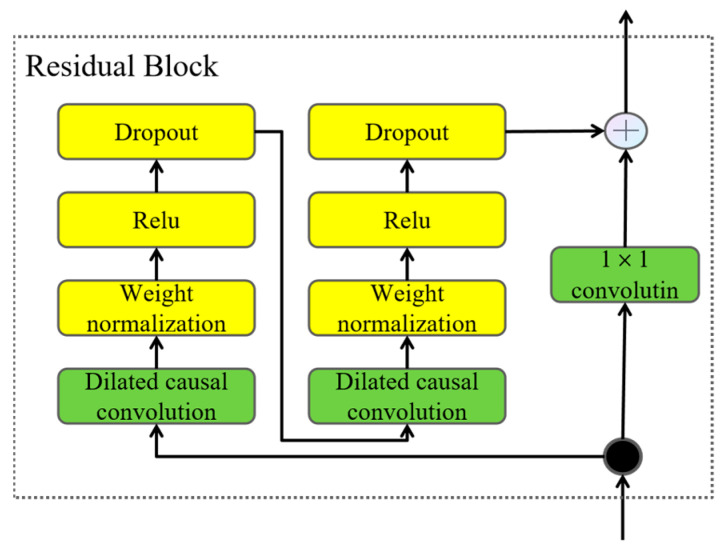
Structure of the temporal convolutional residual block.

**Figure 5 sensors-25-03894-f005:**
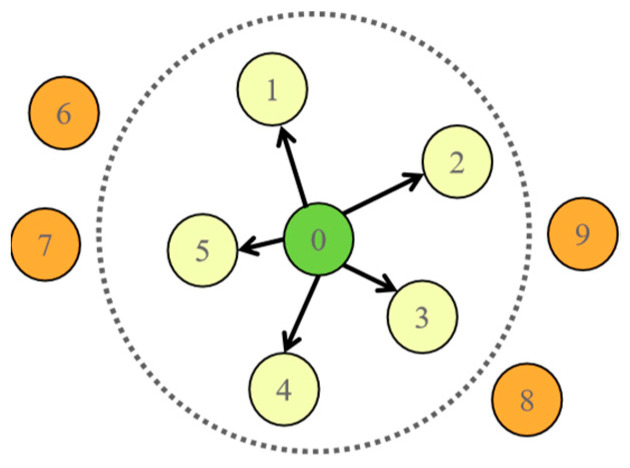
Structure of the KNN graph.

**Figure 6 sensors-25-03894-f006:**
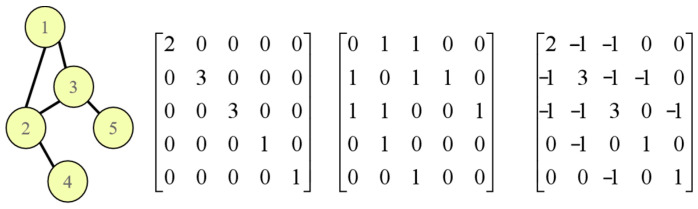
Undirected graph.

**Figure 7 sensors-25-03894-f007:**
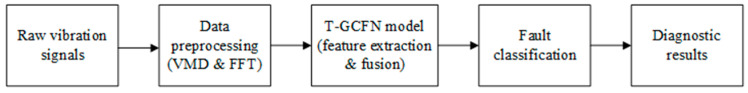
Overall end-to-end workflow of the proposed T-GCFN fault diagnosis method.

**Figure 8 sensors-25-03894-f008:**
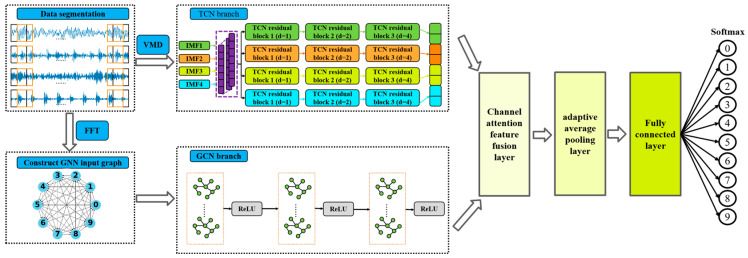
T-GCFN fault diagnosis model architecture.

**Figure 9 sensors-25-03894-f009:**
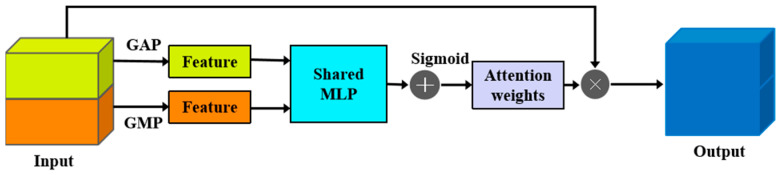
Channel attention feature fusion layer.

**Figure 10 sensors-25-03894-f010:**
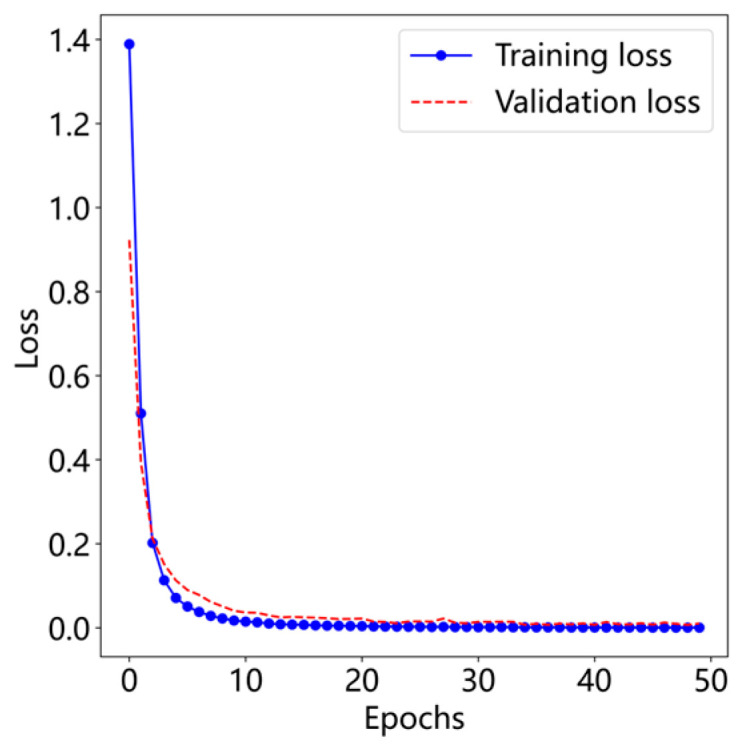
Loss curve.

**Figure 11 sensors-25-03894-f011:**
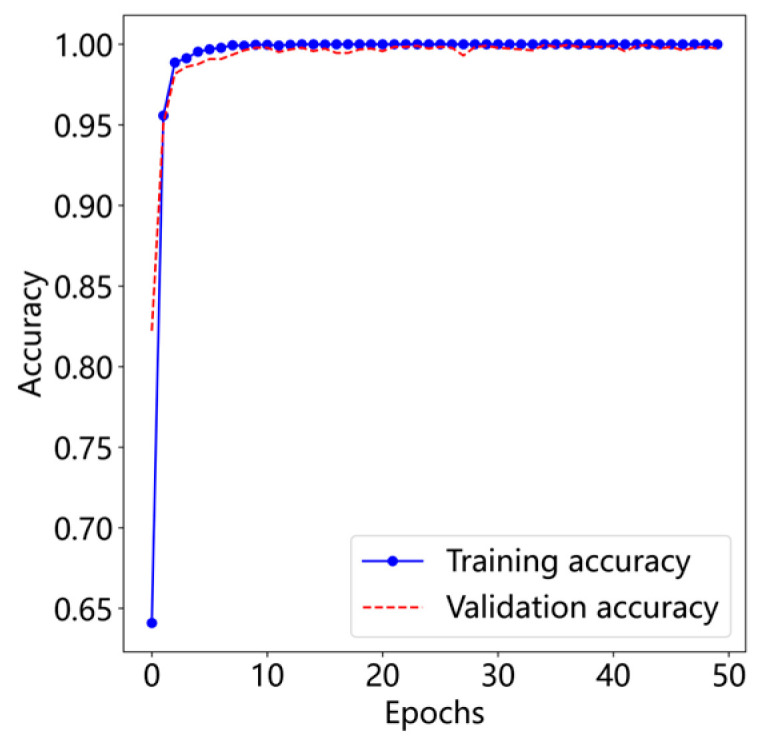
Accuracy curve.

**Figure 12 sensors-25-03894-f012:**
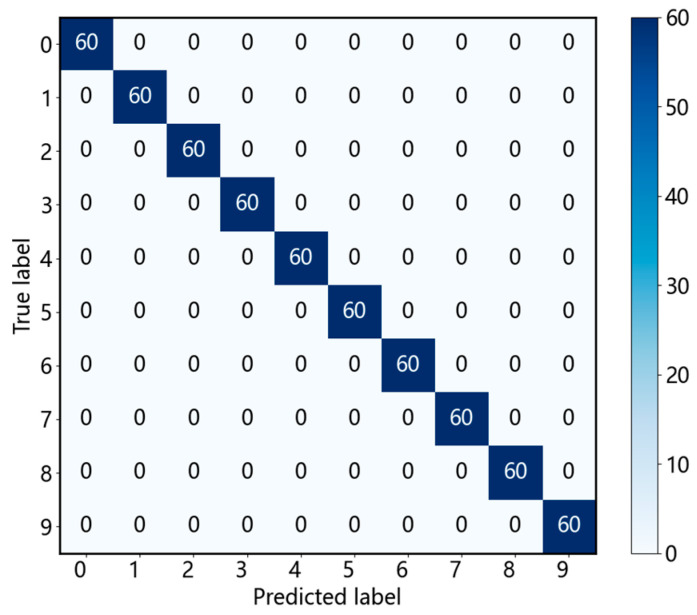
Confusion matrix.

**Figure 13 sensors-25-03894-f013:**
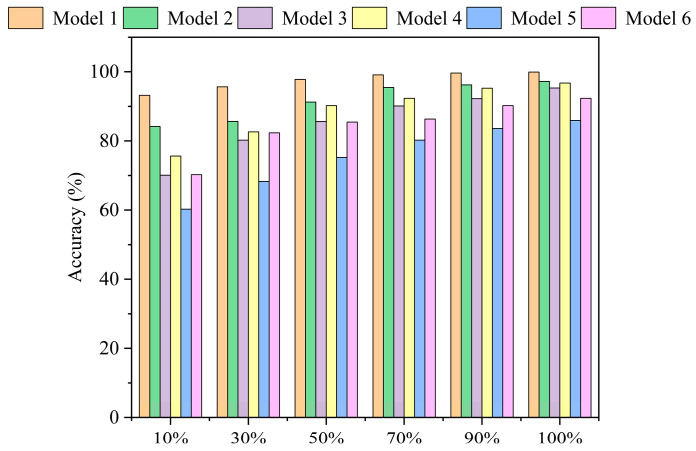
Ablation study results.

**Figure 14 sensors-25-03894-f014:**
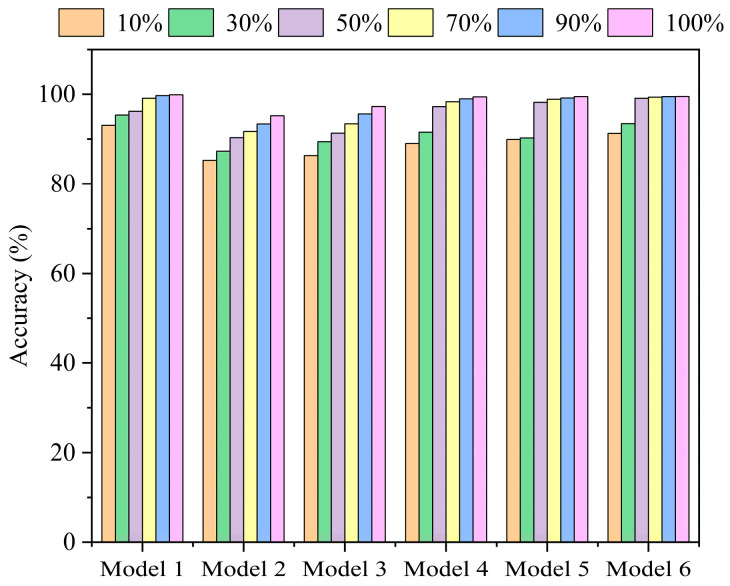
Variable condition experimental results.

**Figure 15 sensors-25-03894-f015:**
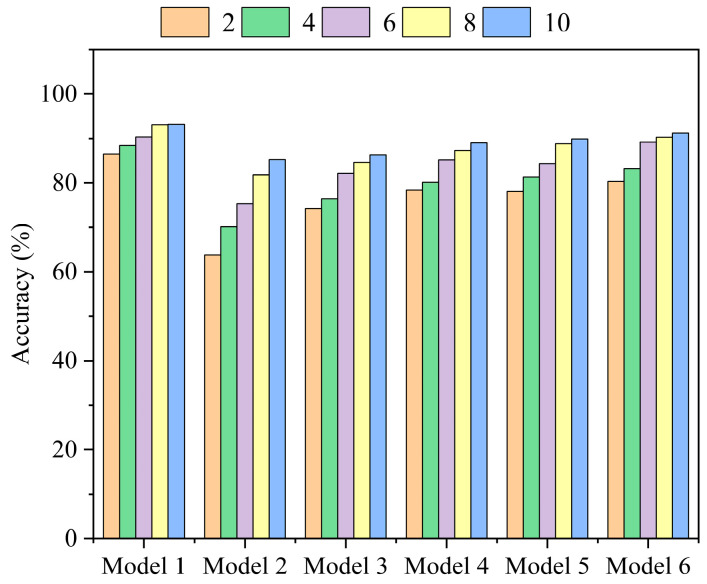
Diagnostic accuracy of different models under various noise levels (10% samples).

**Table 1 sensors-25-03894-t001:** Composition of the CWRU rolling bearing experimental dataset subset (per load condition).

No.	Condition	Defect Size (mm)	Training Samples	Validation Samples	Test Samples	Label
1	Normal	0	280	60	60	0
2	Inner Race Fault	0.1778	280	60	60	1
3	Inner Race Fault	0.3556	280	60	60	2
4	Inner Race Fault	0.5334	280	60	60	3
5	Outer Race Fault	0.1778	280	60	60	4
6	Outer Race Fault	0.3556	280	60	60	5
7	Outer Race Fault	0.5334	280	60	60	6
8	Ball Fault	0.1778	280	60	60	7
9	Ball Fault	0.3556	280	60	60	8
10	Ball Fault	0.5334	280	60	60	9

**Table 2 sensors-25-03894-t002:** Architectural parameters of the proposed T-GCFN model.

Layer Type	Kernel Size	Input Channels	Output Shape
TCN Branch Input	N/A	4	4 × 1024
GCN Branch Input	N/A	10	10 × 512
Initial Conv Layer (TCN)	1 × 128	8	8 × 128
TCN Residual Block(s)	1 × 2	8/16/32	32 × 512
GCN Layers	512 × 512	10	10 × 512
Feature Fusion Module	N/A	60	10 × 512
Global Average Pooling	N/A	60	1 × 512
Output Layer	N/A	10	1 × 10

**Table 3 sensors-25-03894-t003:** The complexity and computational cost of each model.

Model	Average Accuracy (%)	Parameter Quantity (10^5^)	FLOPs (G)	Training Time (s)
Model 1	93.21	4.11	0.12	120
Model 2	84.18	3.12	0.10	102
Model 3	70.1	1.21	0.06	53
Model 4	75.61	1.86	0.07	64
Model 5	60.25	2.14	0.07	78
Model 6	70.25	2.31	0.08	84

**Table 4 sensors-25-03894-t004:** Composition of the Jiangnan University (JNU) rolling bearing experimental dataset (per operating condition).

Dataset	Condition	Training Samples	Validation Samples	Test Samples	Label
A/B/C	Normal	560	120	120	0
	Inner Race Fault (IR)	560	120	120	1
	Rolling Element Fault (RE)	560	120	120	2
	Outer Race Fault (OR)	560	120	120	3

**Table 5 sensors-25-03894-t005:** Diagnostic results on the Jiangnan University (JNU) dataset.

Model	Accuracy (%)			
	A	B	C	Average
T-GCFN	99.58	99.56	99.62	99.59

## Data Availability

The data presented in this study are available in the following repositories: The CWRU dataset is openly available at the Case Western Reserve University Bearing Data Center website at https://engineering.case.edu/bearingdatacenter (accessed on 2 May 2025). The Jiangnan University dataset is available at the CSDN blog at https://blog.csdn.net/qq_41731978/article/details/121733115 (accessed on 2 May 2025).

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
