# Peer review of "Rolling Bearing Fault Diagnosis via Temporal-Graph Convolutional Fusion"

_sensors, 2025, doi:10.3390/s25133894_

Round 1
Reviewer 1 Report
Comments and Suggestions for Authors
The paper is interesting and well-structured, but it can be improved by following the comments:
- Your title is more technical and to make it professional you need to modify it.
- Before section 2.1 you need to briefly describe the importance of "Theoretical Basis".
- Add at least 5 more recent references in your theoretical background
- Please highlight if it is possible that this method can be deployed in real time system monitoring
- In your analysis highlight the key technical innovation by using T-GCF model.
- My suggestion is that you need to include another Figure where can depict the overall data flow by starting from raw signals to final classification.
Reviewer 2 Report
Comments and Suggestions for Authors
The author of this paper proposes a Temporal-Graph Convolutional Fusion (T-GCF) method to solve the problem of insufficient fault features of rolling bearings under the condition of small samples. The author extracted the time-domain branches and frequency-domain branches through variational mode decomposition and fast Fourier transform, processed them by the temporal convolutional network method and the graph convolutional network method, fused the relevant features, and finally carried out the fault identification of rolling bearings. This paper is written smoothly and has a certain degree of logic, indicating the advantages of the time-series - graph convolution fusion method. However, there are some deficiencies and suggestions in the paper as follows:
- There are format issues in lines 112 and 113 of the text.
- The clarity of Figures 7 and 8 in the text is not sufficient. It is suggested that the author improve the clarity of the pictures.
- In line 359 of the text, it is stated that the verification accuracy is close to 100% and the response loss value is close to 0, which does not reflect the precision of the paper. Please correct it to the actual accuracy value and explain that it has a good effect.
- In Section 4.4 of the text, 10%,30%,50%,70%,90%, and 100% of the original training samples are used. However, throughout the entire article, there is no definition of small samples. The number of samples for 10% is 28, so I cannot determine whether 10% is a small sample. Why not 5%, with 14 samples? Even smaller.
- The last two paragraphs of Section 4.4.1 in the text are repeated. Please check the content carefully, author.
- The content of Table 4 in the text is the same as that of Table 5. In the last paragraph, I was unable to obtain the corresponding conclusion stated by the author based on Table 5, and thus had no way to find the source of data such as 99.59%.
- Lines 367 and 368 in the text state that each row represents the True label and each column represents the Predicted label, but they are opposite to the horizontal and vertical axis labels in Figure 11.
- In the process of comparing the six different models in section 4.4.1, the author only compared the accuracy under different models and obtained corresponding conclusions. However, the author did not compare in terms of calculation time, calculation cost, etc., to determine whether the requirements of online monitoring were still met under the condition of obtaining high accuracy.
- In the high-noise experiment in Section 4.4.3, it only explains the relative accuracy of T-GCFN under different dB and different sample sizes, indicating that the accuracy is already good under small samples. However, without comparison with other models, the superiority of T-GCF cannot be proved.
Round 2
Reviewer 1 Report
Comments and Suggestions for Authors
Authors have reflected the changes and recommendation is accepted as it is.
Reviewer 2 Report
Comments and Suggestions for Authors
It can be accpeted.